

# Motor neurons in the escape response circuit of white shrimp (*Litopenaeus setiferus*)

Zen Faulkes

Department of Biology, The University of Texas-Pan American, University Drive, Edinburg, TX, USA

## ABSTRACT

Many decapod crustaceans perform escape tailflips with a neural circuit involving giant interneurons, a specialized fast flexor motor giant (MoG) neuron, populations of larger, less specialized fast flexor motor neurons, and fast extensor motor neurons. These escape-related neurons are well described in crayfish (Reptantia), but not in more basal decapod groups. To clarify the evolution of the escape circuit, I examined the fast flexor and fast extensor motor neurons of white shrimp (*Litopenaeus setiferus*; Dendrobranchiata) using backfilling. In crayfish, the MoGs in each abdominal ganglion are a bilateral pair of separate neurons. In *L. setiferus*, the MoGs have massive, possibly syncytial, cell bodies and fused axons. The non-MoG fast flexor motor neurons and fast extensor motor neurons are generally found in similar locations to where they are found in crayfish, but the number of motor neurons in both the flexor and extensor pools is smaller than in crayfish. The loss of fusion in the MoGs and increased number of fast motor neurons in reptantian decapods may be correlated with an increased reliance on non-giant mediated tailflipping.

Corresponding author
Zen Faulkes, zen.faulkes@utrgv.edu

Decapod crustaceans escape from predators and other sudden stimuli by tailflipping. The neural basis of escape tailflips has been well-studied (*Wine & Krasne, 1972*; *Wine & Krasne, 1982*; *Edwards, Heitler & Krasne, 1999*; *Krasne & Edwards, 2002*; *Faulkes, 2008*), particularly in Louisiana red swamp crayfish (*Procambarus clarkii*). The core of the escape circuit consists of medial giant interneurons (MGs) and lateral giant interneurons (LGs) that drive fast flexor motor neurons, including a specialized fast flexor motor giant (MoG) neuron. Some of these neurons are found in non-decapod crustaceans (*Silvey & Wilson, 1979*), indicating that having this escape circuit is an ancestral condition for the decapods.

Crustacean escape behaviour is an excellent model for studying the evolution of neural circuits. First, the behaviour has an obvious survival value (*Herberholz, Sen & Edwards, 2004*). Second, many of the responsible neurons have no function other than escape. Third, the core neurons responsible for escape should be found in thousands of species. There are over 14,000 decapod crustacean species (*De Grave et al., 2009*), and perhaps 50% may have both MGs and LGs (*Faulkes, 2008*). Indeed, many curious features of the neural circuit cannot be understood without thinking about the evolutionary history of the circuit

(*Edwards, Heitler & Krasne, 1999*; *Krasne & Edwards, 2002*), leading *Krasne & Edwards (2002)* to write, "it may follow that reasonable understanding of the nervous system may be impossible without evolutionary analysis, a most sobering possibility."

Most escape circuit research has been done on crayfish, which belong to Reptantia, a more derived taxon of decapod crustaceans. The more basal decapod taxa, Dendrobranchiata, Caridea, and Stenopodidea (*Dixon, Ahyong & Schram, 2003*; *Porter, Perez-Losada & Crandall, 2005*) are less well-studied, but it is already known that these shrimps and prawns differ in several ways from crayfish. First, crayfish neurons are unmyelinated, but giant interneurons are myelinated in all three non-reptantian taxa (*Holmes, Pumphrey & Young, 1941*; *Heuser & Doggenweiler, 1966*; *Xu & Terakawa, 1999*). Because of the combination of giant interneurons and myelin, shrimp giant axons have the fastest conduction velocity known (*Xu & Terakawa, 1999*). Second, the left and right fast flexor motor giant neurons (MoGs) are separate in crayfish (*Mittenthal & Wine, 1978*), but the MoG axons fuse in caridean shrimp and prawns (*Johnson, 1924*; *Holmes, 1942*; *Friedlander & Levinthal, 1982*). Axonal fusion may promote greater synchrony in muscle activation, which should in turn lead to more powerful tailflips. This should reduce response latency, leading to greater chance of escape. It is surprising that this has been lost in crayfish. Third, at the behavioural level, crayfish giant mediated tailflips are stereotyped and propel the animal in a single plane (*Reichert & Wine, 1983*), but some shrimp can perform a lateral roll during an escape tailflip (*Arnott, Neil & Ansell, 1998*). It is not known how shrimp achieve this, particularly given the bilateral fusion of the MoG axons.

Dendrobranchiate shrimp are the most basal decapod crustacean taxa (*Dixon, Ahyong & Schram, 2003*; *Porter, Perez-Losada & Crandall, 2005*), and thus are in an interesting position for evolutionary studies of the escape response, but little is known about the motor neurons neurons involved in that group. Here, I examine the fast flexor and fast extensor motor systems of white shrimp, *Litopenaeus setiferus*. Some of this work has been presented in abstract (*Faulkes, 2007*).

## METHODS

Live white shrimp, *Litopenaeus setiferus* (Linnaeus, 1767), fished from waters around South Padre Island, Texas, were purchased from commercial seafood stores in Port Isabel, Texas and housed in aquaria. Individuals were anaesthetized by chilling on ice and dissected in physiological saline. The abdominal nerve cord was removed.

Neurons were backfilled (*Pitman, Tweedle & Cohen, 1972*; *Quicke & Brace, 1979*; *Altman & Tyrer, 1980*; *Jones & Page, 1983*). The nerve containing the neurons of interest was placed in a well of petroleum jelly containing 0.3 M solution of either nickel chloride or cobalt chloride, while the remaining tissue was bathed in physiological saline (mM: 410 NaCl, 12.7 KCl, 10.3 $CaCl_2$, 10 $MgCl_2$, and 14 $Na_2SO_4$, 10 tris[hydroxymethyl]aminomethane (Trizma Base); pH adjusted to pH 7.4). The tissue was incubated in a refrigerator for 7–18 h, precipitated with ammonium sulfide or dithiooxamide (a.k.a. rubeanic acid; this term is used hereafter), fixed in 4% formalin in saline, dehydrated with a progressive ethanol series (70% for 10 min, 90% for 10 min, 100% for 10 min, and 100% again for

5 min), and cleared in methyl salicylate. When precipitated with ammonium sulfide, neurons containing cobalt or nickel ions turn black or gray. When precipitated with rubeanic acid, neurons containing cobalt ions turn yellow, and those containing nickel ions turn blue (*Quicke & Brace, 1979*; *Jones & Page, 1983*). Neurons containing some mixture of the two ions turn an intermediate colour, ranging from dark orange to purplish-red (*Quicke & Brace, 1979*; *Jones & Page, 1983*).

The third nerve (N3) was filled 42 times in 30 abdominal ganglia of 14 individuals. The second nerve (N2) was filled 98 times in 61 ganglia of 23 individuals. Abdominal ganglia 1 through 5 were filled, although most fills were of the anterior four ganglia. Because backfills are often incomplete (*Altman & Tyrer, 1980*), the number of cells reported is the maximum number of cells seen across multiple individuals.

Cleared backfills were viewed on an Olympus CX41 microscope, and photographed using an attached Olympus C-5050Zoom digital camera. Images were assembled into final figures using Corel Photo-Paint 12. Some large images were stitched together from multiple photographs.

## RESULTS

### Fast flexor motor neurons

The non-MoG fast flexor cell bodies are found in three clusters (Fig. 1), as in other decapods (*Mittenthal & Wine, 1978*). The flexor medial contralateral (FMC) cluster is contralateral and anterior of the filled N3 (in the terminology of Mittenthal and Wine, "posterior" refers to the position of the axon relative to the cell body). The flexor posterior ipsilateral (FPI) cluster is ipslateral and anterior of the filled N3. The flexor anterior contralateral (FAC) cluster of cell bodies is contralateral and posterior to the filled N3. As in other species (*Mittenthal & Wine, 1978*), there is segmental variation in the number of cell bodies in each ganglion, with the more posterior showing the greatest deviation (Table 1). White shrimp have one or two fewer cell bodies in each cluster than most other decapods examined to date (Table 2). The FMC cell bodies are more widely separated in *L. setiferus* than crayfish, with one anterior of the MoG cell body and near the midline, and the other more posterior and lateral of the MoG cell body. Although this separation means these two cells would not normally be described as being in a "cluster," the FMC in other species is rarely a tight grouping of cell bodies. Fast flexor cell bodies are often pairs or singletons, depending on the number, somewhat separated from other cells in the cluster; e.g., Figs. 3A and 3B in *Espinoza et al. (2006)*; see also *Mittenthal & Wine (1978)*.

The MoG cell bodies in *L. setiferus* are extremely large, have a variegated appearance, irregular shape, and press closely together so that they look like one large mass covering much of the ventral surface of the abdominal ganglion (Fig. 2, Video S1). They are not two widely separate, bilateral, spherical cell bodies reported in caridean prawns (*Holmes, 1942*). In *L. setiferus*, each MoG is ∼300 µm across the ventral surface of the ganglion, and about 100 µm when viewed in the sagittal plane. Other fast flexor motor neurons in *L. setiferus* are ∼50–100 µm in diameter.

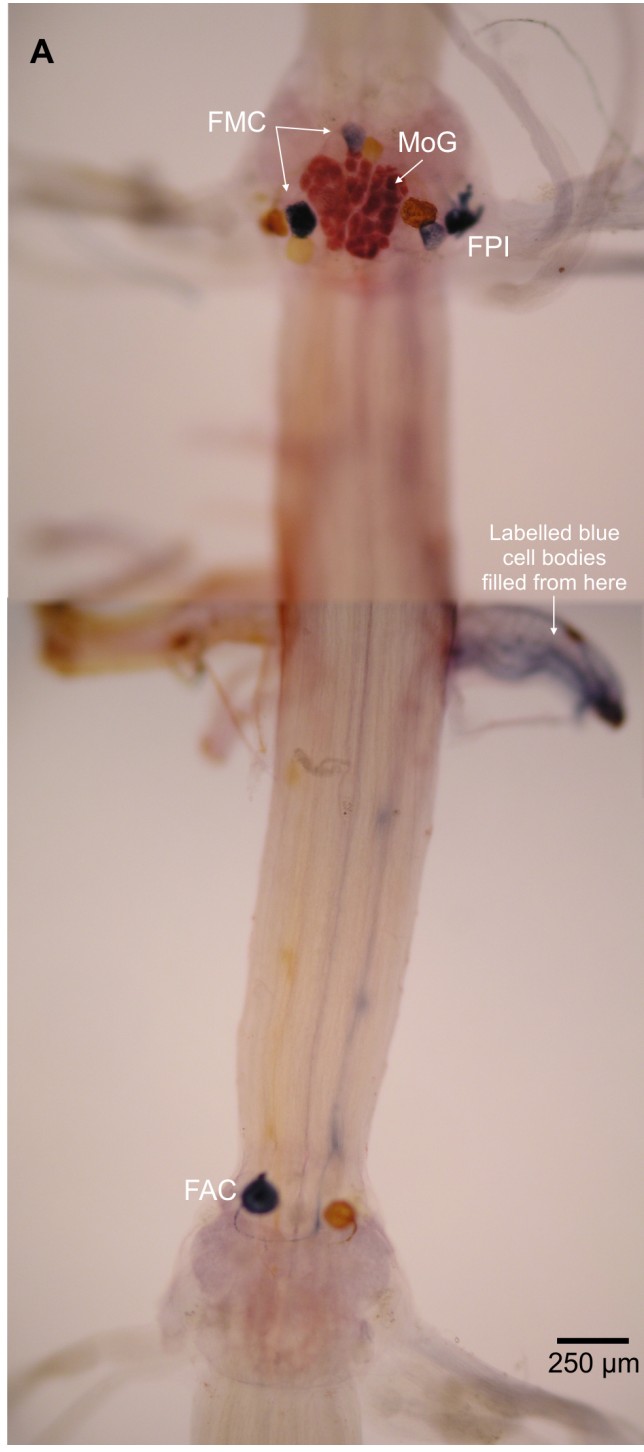

**Figure 1 Fast flexor motor neurons in *L. setiferus*.** Complete fill of all fast flexor neurons, showing both all clusters of motor neuron cell bodies in abdominal ganglia 2 and 3. Cluster labels (FPI, FMC, FAC) shown for cell bodies in blue, filled from nerve shown at right. Bilateral N3 fill of abdominal ganglion 2, with nickel chloride used on N3 shown at right (blue), and cobalt chloride used on N3 shown at left (yellow), precipitated using rubeanic acid. Anterior towards top; ventral view.

**Table 1 Number of fast flexor motor neurons in each abdominal ganglion of *L. setiferus*.**

| Abdominal ganglia(on) | FMC (Non-MoG) | MoG | FPI | FAC |
|---|---|---|---|---|
| A1–4 | 3 | 1 | 3 | 1 |
| A5 | 3 | 1 | 2 | 0 |
| A6 | ? | ? | ? | 0 |

Notes.

FMC, flexor medial contralateral; MoG, motor giant fast flexor motor neuron; FPI, flexor posterior ipsilateral; FAC, flexor anterior contralateral.

**Table 2 Number of fast flexor motor neurons in abdominal ganglion 2 of different species.**

| Species | FMC (non MoG) | MoG | FPI | FAC |
|---|---|---|---|---|
| White shrimp (*Litopenaeus setiferus*) | 2 | 1 | 3 | 1 |
| Spiny lobster (*Panulirus argus*)[a] | 3 | 0 | 4 | 3 |
| Louisiana red swamp crayfish (*Procambarus clarkii*)[b] | 3 | 1 | 4 | 3 |
| American clawed lobster (*Homarus americanus*)[e] | 3 | 1[f] | 4 | 3 |
| Squat lobster (*Galathea strigosa*)[c] | 4 | 0 | 4 | 2 |
| Squat lobster (*Munida quadrispina*)[d] | 3 | 0 | 4 | 0 |

Notes.

FMC, flexor medial contralateral; MoG, motor giant fast flexor motor neuron; FPI, flexor posterior ipsilateral; FAC, flexor anterior contralateral.

[a] *Espinoza et al., 2006*.
[b] *Mittenthal & Wine, 1978*.
[c] *Sillar & Heitler, 1985a*.
[d] *Wilson & Paul, 1987*.
[e] *Otsuka, Kravitz & Potter, 1967*.
[f] See discussion in *Mittenthal & Wine, 1978*.

Two lines of evidence show the MoG axons are fused bilaterally. First, when the N3 of one side is filled with cobalt chloride and the other N3 is filled with nickel chloride and precipitated with rubeanic acid, the MoG cell bodies are dark red, clearly distinct from the blue and yellow of the other cells filled by a single nerve (Figs. 1, 2A and 2B), indicating they are filled with a mixture of both chemicals. Second, a single medial axon is visible in the cord between ganglia, which bifurcates and leaves both the left and right N3 (Fig. 3). Although the MoG cell bodies are pressed so close together to be sometimes indistinguishable as two cell bodies (e.g., Fig. 2B), two axons emerge from the MoG cell bodies (Fig. 2A), project dorsally a short distance within the ganglion before fusing as previously reported (*Johnson, 1924*; *Holmes, 1942*), continuing into the nerve cord as one axon until slightly anterior of the point where N3 exits the body, when it bifurcates and sends a branch both left and right (Figs. 3B and 3C).

The fast flexor motor axons appear slightly "haloed" (Fig. 3C) compared to the smooth axons seen filled though N2 (Fig. 4).

## Fast extensor motor neurons and other N2 neurons

In crayfish, the second nerve (N2) of abdominal ganglia is a mixed nerve that splits some distance from the ganglion. The anterior branch (N2a) contains tactile afferents

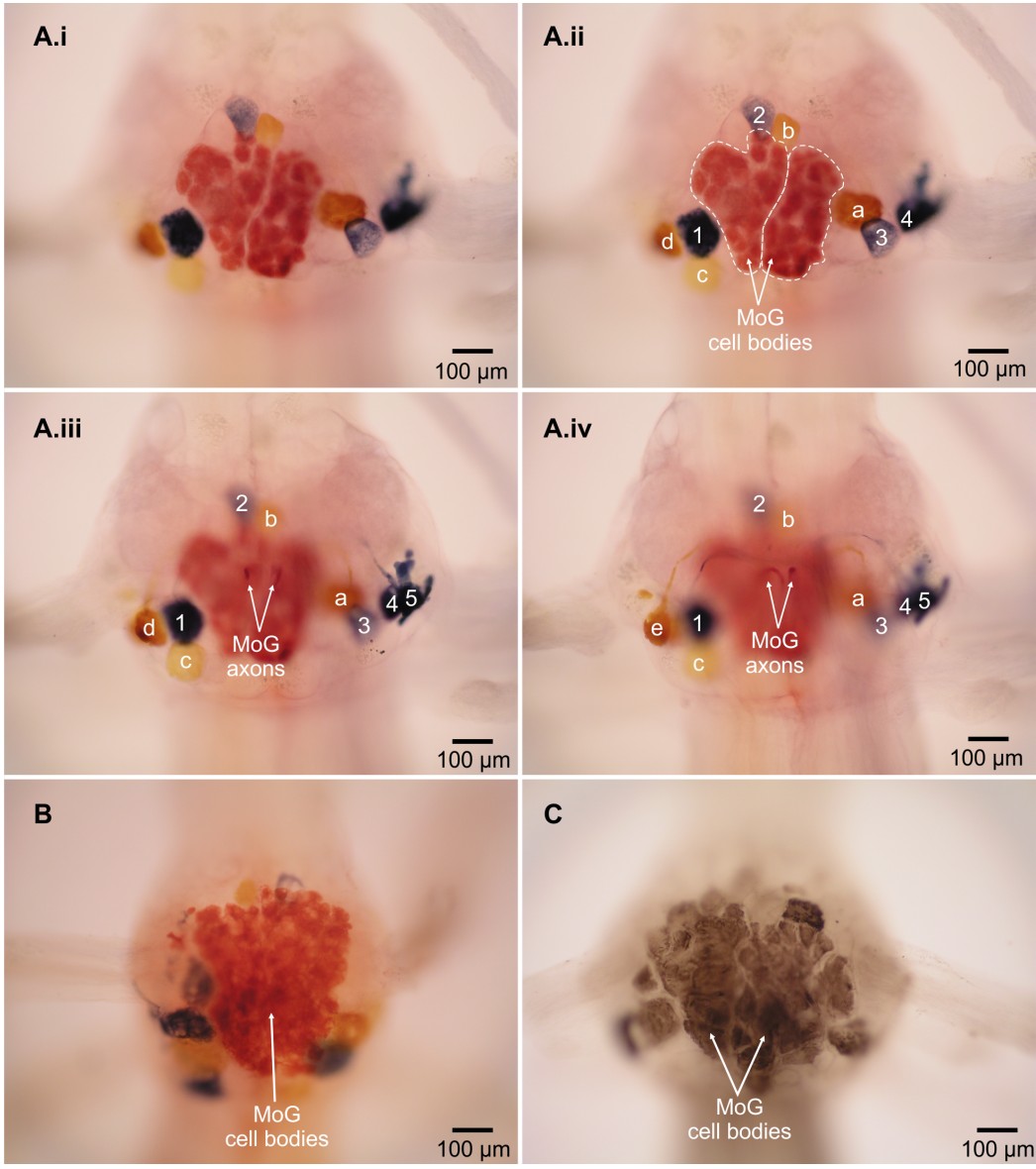

**Figure 2 Motor giant (MoG) cell bodies in *L. setiferus*.** (A) Fast flexor neurons in varying focal planes of abdominal ganglion 2. Same individual in Fig. 1; ganglion is anterior to filled nerve. (i) Unlabeled image showing MoG detail. (ii–iv) MoG cell body (outlined in ii) and axons, FMC cell bodies, and FPI cell bodies not visible in a single focal plane. Letters identify yellow cell bodies of neurons filled with cobalt chloride from left nerve; numbers identify blue cell bodies of neurons filled with nickel chloride from nerve shown at right. (B) MoG structure in abdominal ganglion 1. (C) MoG structure in abdominal ganglion 3. Bilateral fills using cobalt chloride and nickel chloride precipitated with rubeanic acid in (A–B); fills using cobalt chloride precipitated with ammonium sulphide in (C). Anterior toward top; ventral view.

(*Leise, Hall & Mulloney, 1987*). The posterior branch (N2p) contains fast extensor motor neurons (*Treistman & Remler, 1975*; *Drummond & Macmillan, 1998a*), slow extensor motor neurons (*Drummond & Macmillan, 1998b*), and neurons associated with muscle receptor organs (MROs) (*Leise, Hall & Mulloney, 1987*).

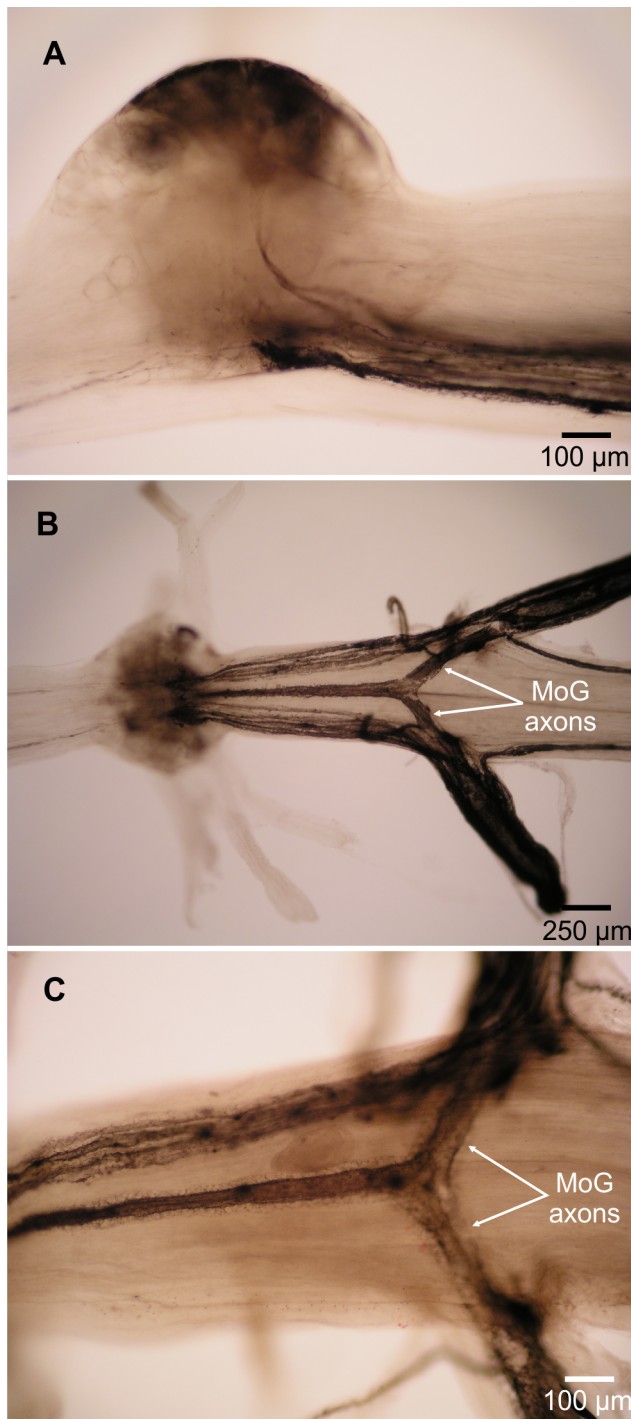

**Figure 3 Motor giant (MoG) axons in *L. setiferus*.** (A) Lateral view of MoG in abdominal ganglion 3, showing axons projecting from ventral cell bodies. (B) Bilateral fill of N3 in abdominal ganglion 3, showing central position of fused MoG axons compared to other fast flexor motor neuron axons. (C) Unilateral fill of abdominal ganglion 1, showing that fill from one side (top of image) results in axon filling that projects out the other, unfilled nerve (bottom). Fills using cobalt chloride precipitated with ammonium sulphide. Anterior towards left of page. Lateral view with dorsal towards top of page in (A), ventral view in (B, C).

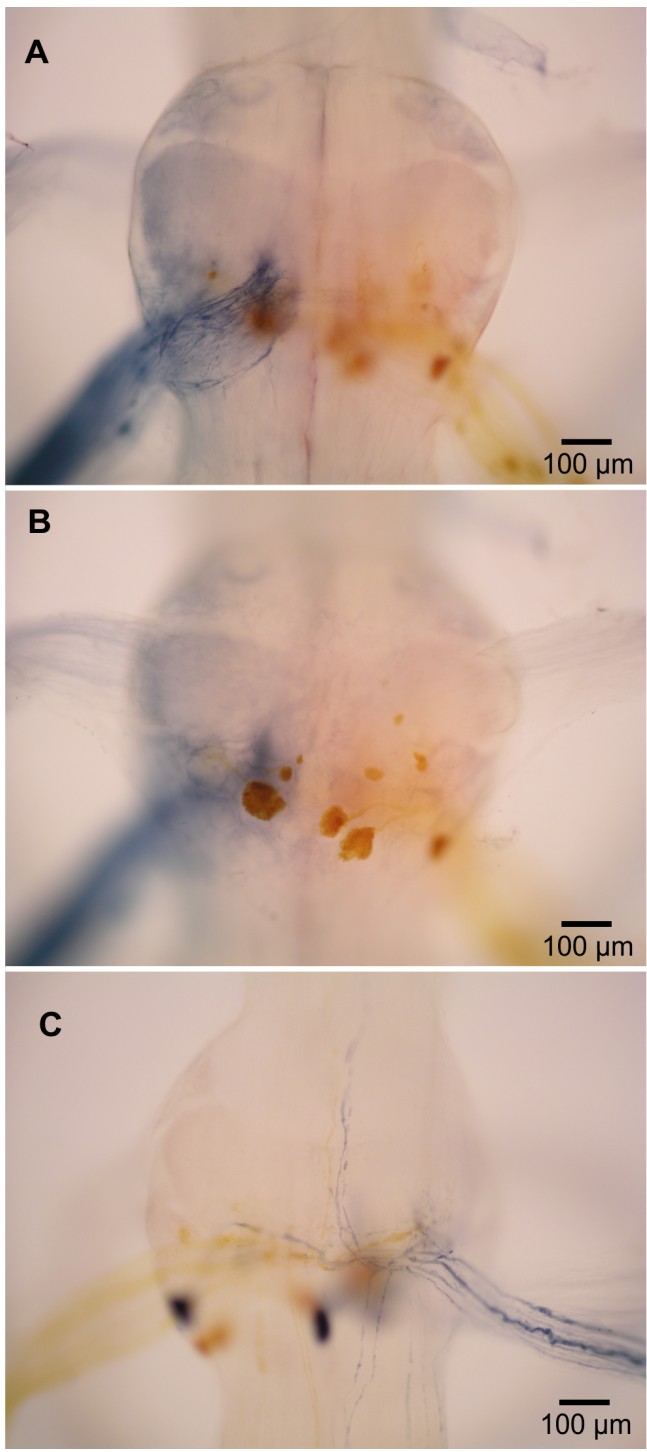

**Figure 4 Extensor-related neurons in *L. setiferus*.** (A, B) Bilateral fill of nerve 2 in abdominal ganglion 1. (A) Putative sensory neurons (blue) filled by anterior branch of nerve 2. (B) Motor neurons (yellow) filled by posterior branch of nerve 2. (C) Muscle receptor organ (MRO) axons (blue) filled through posterior branch of nerve 2 in abdominal ganglion 1. Fills made using cobalt chloride and nickel chloride precipitated with rubeanic acid. Anterior towards top.

**Table 3 Extensor motor neurons and MRO related neurons of different species.**

| Species | Ganglia | Ipsilateral FEMNs | Contralateral FEMNs | Ipsilateral SEMNs | Contralateral SEMNs | Accessory neurons |
|---|---|---|---|---|---|---|
| White shrimp (*Litopenaeus setiferus*) | A1–4 | 3 | 1 | 5? | 1 | 3 |
| Louisiana red swamp crayfish (*Procambarus clarkii*)[a] | A1–4 | 5 | 1 (I) | 5 | 1 | 4 |
| Signal crayfish (*Pacifastacus leniusculus*)[b] | A2–5 | 5 | 3 | 4 | 1 | 4 |
| Australian yabby (*Cherax destructor*)[c] | A3 | 5 | 1 (I) | 5 | 1 | 4 |
| American clawed lobster (*Homarus americanus*)[d] | A1–4 | 3 (EE) | 1 (I) | 4 | 1 | ? |
| Squat lobster (*Galathea strigosa*)[e] | A2 | 4–5 (EE) | 1 (I) | 4 | 1 | 3? |
| Squat lobster (*Munida quadrispina*)[f] | A2–3 | 4 | 1 | 3 | 1 | 3 |

**Notes.**

FEMNs, fast extensor motor neurons; SEMNs, slow extensor motor neurons; EE, extensor excitors; I, inhibitor.

[a] *Treistman & Remler, 1975*; *Wine & Hagiwara, 1977* (but see *Leise, Hall & Mulloney, 1987*, which notes that Wine & Hagiwara misidentified some extensor neurons).

[b] *Leise, Hall & Mulloney, 1987*.

[c] *Drummond & Macmillan, 1998a*; *Drummond & Macmillan, 1998b*.

[d] FEMNs: *Otsuka, Kravitz & Potter, 1967*, SEMNs: *Jones & Page, 1983*.

[e] *Sillar & Heitler, 1985a*; accessory neurons are shown in Figure 9, but the exact number is not mentioned in the text.

[f] *Wallis et al., 1995*; assignment of fast and slow based on examination of Figure 5.

In *L. setiferus*, N2 splits into two branches very near the ganglion, with the anterior branch slightly thicker than the posterior. Fills from N2a revealed many fine processes projecting to the middle of the ganglion and no cell bodies (Fig. 4A). This is a probably a purely sensory branch containing only tactile afferents, as in *Pacifastacus leniusculus* (*Leise, Hall & Mulloney, 1987*).

Many cell bodies fill through N2p (Fig. 4B). Reasonable hypotheses about the identity of cell bodies can be based on their sizes and putative homology with other species (Table 3). Fast extensor motor neurons are usually double or more the diameter of slow extensor motor neurons (*Otsuka, Kravitz & Potter, 1967*; *Wine & Hagiwara, 1977*), although the largest slow extensor motor neurons approach the size of the smallest fast extensor neuron (*Wine & Hagiwara, 1977*; *Drummond & Macmillan, 1998a*; *Drummond & Macmillan, 1998b*). Fills of N2p revealed four large cell bodies located along the posterior margin of the ganglion, one contralateral and three ipsilateral, which are putative fast extensor motor neurons. All abdominal ganglia showed this pattern and no segmental variation was evident. In other decapods, the contralateral cell body is an inhibitory motor neuron and the ipsilateral cell bodies are excitatory motor neurons (*Otsuka, Kravitz & Potter, 1967*; *Treistman & Remler, 1975*; *Wine & Hagiwara, 1977*); the same is likely true in *L. setiferus*.

The other small cell bodies filling through N2p are located in several places. One is found contralateral and posterior, near the putative fast extensor inhibitor; about 3–5 small cell bodies sit along the posterior lateral margin; two are lateral, sitting in the notch between the exit paths of N1 and N2; one small cell body is located anterior of the exit point of N1 (seen in abdominal ganglia 1 and 2; presence in other ganglia unknown); one small cell body is near the exact center of the ganglion.

Two axons from N2p bifurcate near the midline, and send processes both anterior and posterior for unknown distances (Fig. 4C). These are almost certainly axons of the stretch receptors of MROs, which have been described in many species, and are almost always

present as a pair of bifurcating axons (*Sillar & Heitler, 1985a*; *Leise, Hall & Mulloney, 1987*; *Wallis et al., 1995*).

At least three small axons turn posterior and run along the lateral margin of the nerve cord (ganglia 1–3); one exceptionally clear fill in ganglion 1 revealed five such axons. I was unable to fill any cell bodies associated with these axons; fills rarely extended past the exit point of N3. Despite this incomplete picture of their anatomy, these neurons are probably accessory neurons related to the MROs (*Wine & Hagiwara, 1977*; *Leise, Hall & Mulloney, 1987*). There are 4 accessory neurons in abdominal ganglion 2 of *Procambarus clarkii* (*Wine & Hagiwara, 1977*).

The backfilled axons of fast extensor motor neurons do not have the "haloed" appearance of fast flexor motor neurons.

## DISCUSSION

The fast flexor motor giant neurons (MoGs) in *Litopenaeus setiferus* have a structure unlike that reported for any other decapod crustacean. Two aspects of the structure of the MoGs suggest that they may be syncytial cells formed by the fusion of many small neurons. First, they are larger than any other fast flexor motor neurons in this or other species, which are usually ∼100 μm in diameter (*Otsuka, Kravitz & Potter, 1967*; *Mittenthal & Wine, 1978*; *Sillar & Heitler, 1985a*; *Wilson & Paul, 1987*; *Espinoza et al., 2006*). Second, they are not spherical as most other neuron cell bodies are. Third, their variegated appearance suggests they have a different internal structure than other neurons. The hypothesized syncytial structure is reminiscent of the third-order giant neurons in squid stellate ganglia, which are also syncytial cells formed from many cell bodies, and also involved in an escape response (*Young, 1936*; *Young, 1939*). Annelid worms also have syncytial giant neurons (*Nicol, 1948*; *Günther, 1975*) that are involved in escape responses (*Nicol, 1948*; *Günther, 1975*; *O'Gara, Vining & Drewes, 1982*).

Although the size and hypothesized fusion in the MoG cell body is unexpected, it is consistent with the long-known fusion of the MoG axons in other caridean shrimp species (*Johnson, 1924*; *Holmes, 1942*; *Friedlander & Levinthal, 1982*). Given that there are genetic mechanisms to fuse the MoG axons during development (*Friedlander & Levinthal, 1982*), the same mechanisms could be used to fuse cell bodies. Reduction of fusion appears to be an evolutionary trend in the decapods, starting with hypothesized fusion of MoG cell bodies (this study) and axons in dendrobranchiates (this study; *Xu & Terakawa, 1999*), to fusion of the MoG axons only in carideans (*Holmes, 1942*), to no MoG fusion in reptantians (*Mittenthal & Wine, 1978*). A prediction of this fused cell body hypothesis is that the MoG cell bodies would contain multiple nuclei. Because backfills rarely reveal any subcellular structure, other techniques, such as thin sectioning and staining for higher resolution microscopy, will be needed to test this hypothesis.

The remaining fast flexor and fast extensor motor neurons appear to be found in homologous positions to the better studied reptantian species. In almost every case, there are fewer cell bodies in *L. setiferus* than in the homologous groups of neurons in most reptantian decapods. If other non-reptantian species have similarly small numbers

of motor neurons, it would suggest that duplication of fast abdominal motor neurons occurred during decapod evolution. The loss or reduction of the massive MoG cell bodies may be correlated with the increased number of fast abdominal motor neurons in reptantians: the amount of ganglionic volume consumed by the MoGs may have constrained the addition of any new fast motor neurons.

The "haloed" appearance of the fast flexors, but not fast extensors, may be indicative of myelination. In *Palaemon serratus*, the MoG axons are myelinated in the periphery, and references to other axons of similar size being myelinated suggest the other fast flexor motor neurons are also myelinated (*Holmes, 1942*).

The smaller number of fast abdominal neurons might indicate that shrimp have less fine grained control over the fast flexor motor muscles than crayfish. In many reptantian crustaceans, the fast flexor muscles are used in two distinct forms of tailflipping. The MoGs and other fast flexor neurons are used in single stereotyped escape tailflips triggered by giant interneurons. The MoGs are not involved in repetitive variable tailflipping, which is controlled by an undescribed system of non-giant interneurons (*Reichert, Wine & Hagiwara, 1981*; *Reichert & Wine, 1983*; *Sillar & Heitler, 1985b*; *Wilson & Paul, 1987*; *Faulkes, 2004*); non-giant tailflipping would be generated by the remaining pool of fast flexor motor neurons. A larger pool of motor neurons may allow for some of the fine control necessary for such variability. Previously, I suggested that non-giant tailflipping originated at the base of the decapod clade (*Faulkes, 2008*), but I did not express this hypothesis as tentatively as it should have been. It is not known if non-reptantian decapods have variable, non-giant tailflipping behaviour like many reptantians do. Indeed, the myelination of the entire population of fast flexors and the axonal fusion of the MoGs point to a circuit specialized for explosive starts. It may be that tailflipping cannot occur without activity of the giant neurons in shrimp, and that a small pool of neurons is sufficient to generate shrimps' more consistently explosive tailflips. Alternately, the variation in motor neuron number may be trivial and have little functional impact, because crustacean muscles generally have sparse polyneuronal innervation. In *Munida quadrispina*, the FAC cluster of fast flexor motor neurons was lost with no visible change in tailflipping behaviour (*Wilson & Paul, 1987*).

### Funding
The author declares there was no funding for this work.

### Competing Interests
The author declares there are no competing interests.

### Author Contributions
- Zen Faulkes conceived and designed the experiments, performed the experiments, analyzed the data, contributed reagents/materials/analysis tools, wrote the paper, prepared figures and/or tables, reviewed drafts of the paper, created figures.

## Supplemental Information

Supplemental information for this article can be found online at http://dx.doi.org/10.7717/peerj.1112#supplemental-information.

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
