# Peer review of "Motor neurons in the escape response circuit of white shrimp (Litopenaeus setiferus)"

_PeerJ, doi:10.7717/peerj.1112_

## Round 0.1 · original submission · Major Revisions

· Academic Editor

Major Revisions

The reviewers agree on the value of the work, yet raise some concerns about the extent to which the data presented justify some of the claims made in the manuscript. It is my impression that the additions and changes that the reviewers suggest are feasible and would significantly improve the quality of the manuscript. I look forward to receiving a revised copy of the work.

·

Basic reporting

the basic reporting criteria are all perfectly met; I attach an annotated pdf to draw the authors' attention to a few typos and a few minor suggestions for improvement.

Experimental design

All experimental design criteria are met.

Validity of the findings

All reported findings are valid, particularly according to the journal's criteria.

Additional comments

I attach an annotated pdf to draw your attention to a few typos and a few minor suggestions for improvement. None of these suggestions are essential for publication! I ticked the "publish after minor revision" button only to alert you to those few typos.

Reviewer 2 ·

Basic reporting

There are some typographical/grammatical errors:
L 80 “press closely together so closely”
Legend fig 1. “… both all…”
L 223. No journal provided in reference.

There does not appear to be any reference to the figures within the Results section. Unless this is a journal standard, then I think figures should definitely be referenced as results are described.

The research background and motivation are clear.

Experimental design

The aim is to describe the abdominal flexor and extensor motor pool in a basal decapod group that has not been previously studied. The author uses cobalt/nickel backfilling, which is adequate for an initial description, but leaves open some of the intriguing questions raised by the findings.

Validity of the findings

The main findings regarding the numbers and locations of standard flexor and extensor motorneurons seem sound. The most interesting findings are relative to the MoG, and here there is some speculation. The conclusion that the bilateral axons are fused is entirely plausible given what is known in other species, but is certainly not demonstrated beyond doubt in the data presented. The magnification in Fig. 2D is too low to be sure that the images are not simply paired axons closely apposed. It would have been better given the technique employed, to back-fill just one nerve root rather than the bilateral pair, and to look for dye-coupling and the exit of a large MoG axon in the unfilled root.
The other main suggestion is that the cell body itself may be syncytial. This is certainly consistent with the "bunch of grapes" appearance of the ganglionic stain in Fig 2B. However, so far as I know this is a unique suggestion (in decapod crustaceans), and as such, personally I would prefer to see it enter the literature with proper evidential backing, rather than as a suggestion based on the not-absolutely-clear appearance of wholemount back-fills. Even something as simple as cross-section micrographs from within the ganglion could add considerable weight to the idea. However, the author does identify the idea as being a suggestion rather than a substantiated claim, and so if I understand it correctly it falls within the acceptable policy of the journal.

Additional comments

I honestly think there is some fascinating potential findings here, but I also think that the article could be seriously improved with some additional work to convert the speculation into substantiated claims. However, consideration of the degree of advance contained within the article is excluded as a criterion for review, and as such the data on which definite claims are based are sound.

Reviewer 3 ·

Basic reporting

1. This paper describes the structure of the major motor neuronal elements of the abdominal tail flip escape circuitry of the white shrimp and compares it to homologous neurons in the crayfish and other decapod crustaceans. The paper could be a useful contribution, but suffers from many flaws, as described below. Most of these flaws could be easily corrected by changes to the text and the addition of images needed to support the paper’s claims.

Experimental design

The experimental design is acceptable, except in the analysis of cell body counts as described below.

Validity of the findings

2. A major point of the paper concerns the possible fusion of the MoG motor neuron axons in each segmental connective. Whereas Holmes (1942) presented convincing evidence for axonal fusion between bilateral MoG motor neurons in the prawn Leander serratus, no similar evidence is presented here for the white shrimp. The bilateral back-fills of nerves 3 shown in Fig. 2D show that the left and right MoG axons may fuse along the ventral cord midline, but they may also be separate cells. Convincing evidence may be available in the preparation of Fig. 1, where different metals are used to fill opposing 3rd nerves. If MoG were a single fused cell, one might expect that a fill of one root with cobalt chloride would reveal a cobalt-stained axon in the opposing root, but that is not apparent, although the opposing root is out of focus. Moreover, the pair of large MoG cell bodies are not apparent among the cobalt-stained cell bodies in that figure. To sum, much more convincing evidence needs to be presented that the MoGs in each segment are fused.
3. The figures are not referenced in the text, which makes the text difficult to understand. It frequently appears to refer to elements in the figures, but which elements those are is unclear. The lack of figure references means that many statements of fact in the paper are unsupported. All statements of fact should be supported either by a figure or table reference, with the statement “not shown”, or by a citation to another work.
4. Line 79-80: The description of the MoG cell bodies does not reference Fig. 1 or 2 and is not clearly supported by either of them. Given that the focus of the paper is largely on these cells, it would be useful to have an image of them that supports the description.
5. Fig. 1: Explain the cell numbering in B-D. Do the numbers and letters refer to MNs exiting in opposing 3rd roots? Which root has numbered somata and which has lettered somata?
6. Discussion. Lines 152-154. The statements about the structure of the MoGs are not supported by any of the results presented in the paper, including Figs. 1 and 2. What about the structure of the MoGs “..suggests that the MoGs may be syncytial cells formed by the fusion of many small neurons”? The only statement in the Results section that addresses this is in lines 79-84, where the MoG cell bodies are described as having a “variegated appearance, irregular shape and press closely together so closely that they look like one large mass covering much of the ventral surface of the surrounding ganglion”. As stated in 4 above, there is no image of this presented, and no conclusion is drawn from this until the “suggestion” presented in the Discussion.
7. It is unclear from the table legends or from the text whether the neuronal counts presented in the tables are averages from different preparations (14 animals were stained) or counts from a single preparation. If they are averages, what is the variation in the numbers across preparations? Given the variability in staining, what criteria were used to include/exclude cells?

---

## Round 0.2 · accepted · Accept

· Academic Editor

Accept

Unfortunately, the reviewers who had issues of substance with the previous version were not available to review your new version. I have therefore consulted with Claudio Lazzari from the editorial board, who agrees with me in that the issues raised by the reviewers have been adequately addressed. He did raise, however, concerns about backfilling used as a single criterion for the hypothesis of syncytial neurons. I would therefore ask you to consider adding one or two sentences to make readers aware of the limitations (and possible artifacts) of the technique and how these might impact your conclusions.

Reviewer 4 ·

Basic reporting

.

Experimental design

.

Validity of the findings

.

Additional comments

After reading through the revised manuscript by Zen Faulkes (Motor neurons in the escape response circuit of white shrimp (Litopenaeus setiferus)), I see no need for another revision and would suggest to proceed with publication.